

# Dopamine and eye movement control in Parkinson's disease: deficits in corollary discharge signals?

Henry Railo[1,2,3], Henri Olkoniemi[3], Enni Eeronheimo[3], Oona Pääkkönen[3], Juho Joutsa[4,5,6,7] and Valtteri Kaasinen[2,6,7]

[1] Department of Clinical Neurophysiology, University of Turku, Turku, Finland
[2] Turku Brain and Mind Centre, University of Turku, Turku, Finland
[3] Department of Psychology, University of Turku, Turku, Finland
[4] Athinoula A. Martinos Center for Biomedical Imaging, Harvard Medical School, Charlestown, MA, USA
[5] Berenson-Allen Center for Noninvasive Brain Stimulation, Harvard Medical School, Charlestown, MA, USA
[6] Department of Neurology, University of Turku, Turku, Finland
[7] Division of Clinical Neurosciences, Turku University Hospital, Turku, Finland

Corresponding author
Henry Railo, hmrail@utu.fi

## ABSTRACT

Movement in Parkinson's disease (PD) is fragmented, and the patients depend on visual information in their behavior. This suggests that the patients may have deficits in internally monitoring their own movements. Internal monitoring of movements is assumed to rely on corollary discharge signals that enable the brain to predict the sensory consequences of actions. We studied early-stage PD patients ($N = 14$), and age-matched healthy control participants ($N = 14$) to examine whether PD patients reveal deficits in updating their sensory representations after eye movements. The participants performed a double-saccade task where, in order to accurately fixate a second target, the participant must correct for the displacement caused by the first saccade. In line with previous reports, the patients had difficulties in fixating the second target when the eye movement was performed without visual guidance. Furthermore, the patients had difficulties in taking into account the error in the first saccade when making a saccade toward the second target, especially when eye movements were made toward the side with dominant motor symptoms. Across PD patients, the impairments in saccadic eye movements correlated with the integrity of the dopaminergic system as measured with [$^{123}$I]FP-CIT SPECT: Patients with lower striatal (caudate, anterior putamen, and posterior putamen) dopamine transporter binding made larger errors in saccades. This effect was strongest when patients made memory-guided saccades toward the second target. Our results provide tentative evidence that the motor deficits in PD may be partly due to deficits in internal monitoring of movements.

## INTRODUCTION

Parkinson's disease (PD) leads to hypometric and fragmented eye movements. These deficits in are most prominent when the eye movements cannot be guided by external

visual information (*Terao et al., 2011*; *Rieger et al., 2008*; *Hodgson et al., 1999*; *Kimmig et al., 2002*; *Blekher et al., 2009*). In general, patients with PD are known to depend on visual feedback in controlling their movements more than healthy individuals (*Klockgether et al., 1995*; *Jobst et al., 1997*; *Glicksterin & Stein, 1991*). This suggests that PD patients may have deficits in monitoring their own movements, a process assumed to depend on corollary discharge (CD) signals that relay a copy of the motor command to sensory areas (*Wurtz, 2008*). The present study aimed to examine if PD patients show deficits in monitoring the eye movements they have performed. A secondary objective was to assess if the impairments in saccadic eye movements correlates with the integrity of the dopaminergic system as measured with dopamine transporter (DAT) imaging in human PD patients.

Saccadic eye movements are triggered by neurons in the superior colliculus (SC). Saccade-related activity in SC is assumed to be triggered by activation of the caudate nucleus, which releases the SC from tonic inhibition exerted by the substantia nigra pars reticulata (*Hikosaka, Takikawa & Kawagoe, 2000*). The basal ganglia are thus assumed to play the role of a gatekeeper, securing that the driving input from cortical areas that guide the voluntary initiation of eye movements do not lead to a chaotic cascade of saccades (*Hikosaka, Takikawa & Kawagoe, 2000*). Put differently, the basal ganglia is in a position to select the appropriate action given incoming sensory information and task demands (*Shires, Joshi & Basso, 2010*).

One of the defining features of PD is the degeneration of dopaminergic neurons in the substantia nigra pars compacta (*Kordower et al., 2013*), which leads to excessive inhibition in the basal ganglia and the thalamus, and results in decreased motor output (*Albin, Young & Penney, 1989*; *DeLong & Wichmann, 2007*; *Calabresi et al., 2014*). In PD patients this results in hypometric saccadic eye movements, and this defect becomes more pronounced in conditions where the saccades are memory-guided (as opposed to visually-guided), when compared to healthy controls (*Kimmig et al., 2002*; *Blekher et al., 2009*; *Crawford, Henderson & Kennard, 1989*; *Terao et al., 2011*). That specifically memory-guided saccades are impaired in PD is consistent with the observation that the dopamine system supports the accurate maintenance of memory traces by modulating neural activity in prefrontal cortex (*Floresco & Phillips, 2001*; *Landau et al., 2009*). During the execution of voluntary saccades, PD patients display lack of activation especially in frontal cortical areas (*Rieger et al., 2008*). Striatum and prefrontal cortex could also mediate action monitoring as activity in these areas have been observed to correlate with information about recent actions and goals (*Tsujimoto & Postle, 2012*; *Genovesio & Ferraina, 2014*; *Kim, Lee & Jung, 2013*).

Direct evidence concerning the effects of dopamine on saccadic eye movements is scarce in human subjects. *Hotson, Langston & Langston (1986)* showed that saccades become infrequent and hypometric in MPTP-induced parkinsonism. Similar observations have been in monkeys (*Brooks, Fuchs & Finocchio, 1986*; *Schultz et al., 1989*; *Kato et al., 1995*). Impairments in memory-guided saccades have been observed also in humans with caudate lesions (*Matsumura, Fukasawa & Kojima, 1996*). The present study examined whether the deficits in making memory-guided saccades correlates with dopaminergic

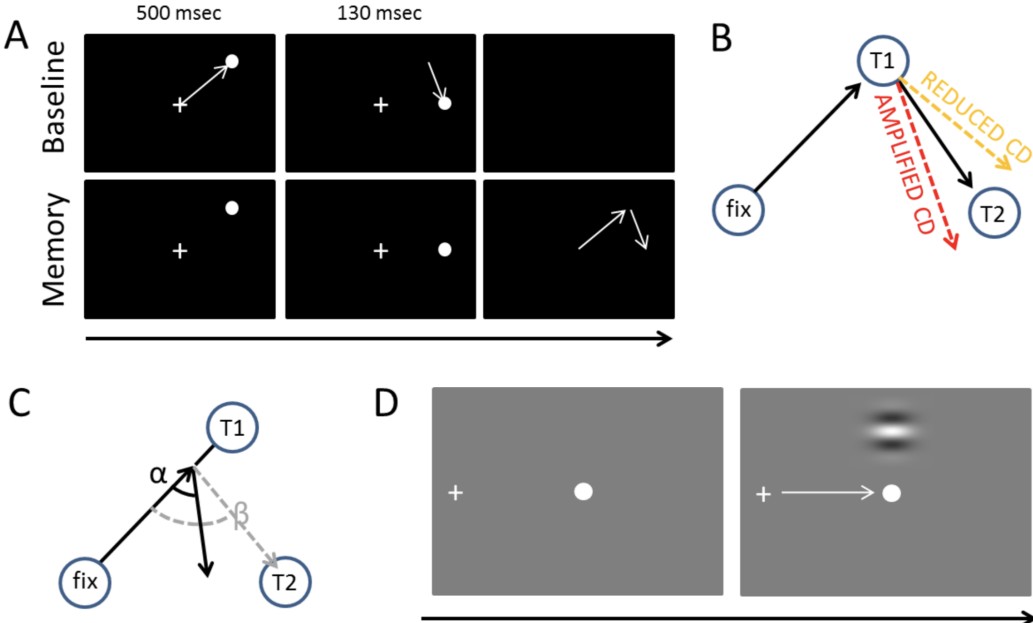

**Figure 1 Tasks and measures.** (A) The double-saccade task. In the baseline condition the participants made two sequential visually-guided saccades (white arrows) to targets. In the Memory condition, the participants made saccades to memorized target locations. (B) Hypothesis concerning the CD. If CD is diminished relative to the strength of the motor command, the participants fail to fully take into account the first eye movement they have performed—hence, their second saccades are biased outward (orange dashed line). Amplified CD (relative to the strength of the motor command) produces the opposite pattern of results: the second saccade is biased inward (red dashed line). (C) The efficacy of sensorimotor control can also be examined on a trial-by-trial basis by measuring how well the participants take into account errors in the first saccade (*Joiner, FitzGibbon & Wurtz, 2010*). In the example, the participant's first saccade is hypometric. The gray dashed line shows what would be the optimal amplitude (length of arrow) and angle ($\beta$) of the second saccade to compensate for the error in the first saccade. The second black arrow shows an example where the angle ($\alpha$) and amplitude of the second saccade is sub-optimal (i.e., participant overestimates the length of the first saccade, for example, due to amplified CD relative to the actual motor output). (D) Visual detection thresholds were measured (2-alternative forced-choice task) by presenting a Gabor patch in the top or bottom half of the screen during fixation, saccade, or right after saccade. An adaptive staircase method was used to adjust the contrast of the patch to find the 81% detection threshold.

deficiency in human PD patients. We also hypothesized that the saccadic deficits in PD may reflect an impairment in predicting and monitoring the outcomes of motor programs.

To trigger spatially accurate saccades, and to stitch together a stable visual percept from the samples provided by individual eye fixations, the brain must predict the sensory consequences of the eye movements (*Wurtz, 2008*). In order to examine how well PD patients can internally monitor the eye movements they make, we employed a *double-saccade task* which requires the participants to take into account the displacement caused by a first saccade to accurately perform a second saccade (Fig. 1) (*Wurtz, 2008*; *Ostendorf, Liebermann & Ploner, 2010*; *Thakkar et al., 2015*). CD signals that relay a copy of the motor command to sensory areas via medial thalamus are crucial for this spatial remapping (*Sommer & Wurtz, 2008*; *Wurtz, 2008*). The dopaminergic deficit in PD results in abnormal functioning of the structures that are assumed to relay the CD to

cortex (*DeLong & Wichmann, 2007*; *Albin, Young & Penney, 1989*; *Pifl, Kish & Hornykiewicz, 2012*).

The CD signal can be seen as a part of a more general mechanism that integrates bottom-up sensory information and contextual information. This is clearly formalized in the predictive coding framework which assumes that the brain is constantly attempting to predict (model) incoming sensory information based on previous experiences (*Hohwy, 2013*; *Clark, 2016*). The CD signal conveys a prediction about the sensory consequences of a movement. According to this framework, during (eye) movements the brain compares its current model of the world (predictions) against incoming sensory information (*Friston et al., 2012a*). Saccades (or any other actions) that are guided by external sensory stimuli are predictions which aim to minimize the discrepancy between the brain's model of the world (predictions), and incoming sensory information (prediction error) (*Friston et al., 2010*). Voluntarily triggered saccades are predictions about the sensory consequences of those movements (*Friston et al., 2010*). *Friston et al. (2012b)* have hypothesized, and using dopamine depletion simulations provided evidence, that the dopaminergic system mediates sensorimotor integration by adjusting the how much the brain relies on the predictions. Specifically, they have proposed that the dopaminergic deficiency in PD causes the patients to rely more on their internal predictions (i.e., their model of the world) than bottom-up sensory information (prediction errors such as proprioceptive feedback or visual input) (*Friston et al., 2012b*).

Defects in CD have been observed in neurological patients with thalamic lesions (*Bellebaum et al., 2005*; *Ostendorf, Liebermann & Ploner, 2010*). Using the double-saccade task, *Thakkar et al. (2015)* observed that in schizophrenia patients the errors in the angle of the second saccade suggested reduced CD relative to motor signals (although this effect was only observed for one of the two possible saccade target arrangements). As schizophrenia has been associated with hyperactive striatal dopamine neurotransmission (*Howes & Kapur, 2009*), this raises the question whether PD patients—who characteristically have nigrostriatal dopaminergic defects (*Kordower et al., 2013*)—display a reverse effect (i.e., amplified CD relative to motor commands). Previous studies have not tested whether the integrity of the dopaminergic system correlates with performance in the double-saccade task. While dopamine is not per se assumed to relay the CD information, it is assumed to contribute to internal monitoring of movements by mediating the integration of bottom-up sensory information and contextual information such as previously performed movements (e.g., CD) (*Friston et al., 2012b*). The two possible outcomes of CD deficits (reduced vs. amplified CD relative to the motor signals) are displayed in Fig. 1B. As shown in Fig. 1C, CD also allows for taking into account variation in the endpoint of the first saccade on a trial-to-trial basis to produce accurate saccades to the second target (*Joiner, FitzGibbon & Wurtz, 2010*; *Munuera et al., 2009*). Deficits in the trial-by-trial monitoring of saccades could produce the fragmented, multi-step saccade pattern which is characteristic of PD patients (*Kimmig et al., 2002*).

Deficits in sensorimotor integration could lead to problems in visual stability and contribute to the motor symptoms of PD as the patient's internal model of the world does not accurately reflect the state of the world. Abnormal CD could also contribute to visual

symptoms observed in PD (*Weil et al., 2016*), such as hallucinations (*Diederich et al., 2014*; *Thakkar, Diwadkar & Rolfs, 2017*). Hence, the present study included a short questionnaire that probed for possible deficits in the experienced visual stability in PD. We also measured visual detection threshold of the participants during fixation and saccades because CD-type "extraretinal" signals have been hypothesized to underlie the suppression of visual sensitivity during saccades (i.e., saccadic suppression) (*Ross et al., 2001*).

## METHODS

### Participants

Eight female ($M_{age}$ = 60, range: 45–72 years) and six male ($M_{age}$ = 62, range: 44–74 years) PD patients (13 right-handed and one ambidextrous) and seven female ($M_{age}$ = 60, range: 60–74 years) and seven male ($M_{age}$ = 67, range: 44–78 years) right-handed neurologically healthy control participants took part in the experiment. PD patients were diagnosed using the UK Brain Bank criteria, all diagnoses were supported by marked striatal defects in DAT SPECT and none of the patients presented features suggestive of atypical parkinsonism. All participants had normal or corrected-to-normal vision. Three control participants had to be excluded from the analysis due to technical problems during testing. In most cases the control participants were the spouses of the PD patients. The duration of motor symptoms in PD patients was on average 24 months (range: 3–72 months), and the Hoehn–Yahr stages of the patients were as follows: Six patients at stage I, seven patients at stage II, and one patient at stage III. The patients had a mean motor MDS-UPDRS score of 37 (range 11–95, SD = 20). In eight patients the motor symptoms were primarily on the left side, in four patients in the right side, and two patients had nearly symmetrical motor symptoms. None of the patients had depression (mean BDI = 3.8, range 1–12, SD = 2.8), and none showed signs of abnormal cognitive abilities as measured by the Mini-Mental State Examination (mean 27.6, range 26–30). Only PD patients completed the BDI and MMSE examination (during SPECT imaging, see below).

A total of 11 patients underwent a voluntary break in medication before the experiment and did not take medication on the morning of the test session. One patient did not have any medication, and two participants did not go through a break in medication. Because the majority of the patients were on 12-h medication break, these patients can be considered an OFF phase. A voluntary break in medication was encouraged because we assumed that PD medication may counteract the possible deficits in CD. Three patients were on levodopa, nine patients were treated with MAO-B inhibitors and/or dopamine agonists and medication was unknown for one patient.

The experiment was approved by the Ethics committee of the hospital district of South–West Finland (ETMK 18/1801/2016), and a written informed consent was obtained prior the study. The study was conducted according to the principles of the Declaration of Helsinki.

### Tasks, stimuli and procedure

The present study included three parts. First, the participants responded to a short questionnaire about their subjective experience of visual stability. Second, the participants

completed the double-saccade task. Third, the strength of saccadic suppression in the participants was evaluated by measuring visual thresholds during saccades and fixation.

In the questionnaire the participants were asked if they had perceived motion, experienced vertigo, or had feelings that the world has "changed position" during saccadic or smooth pursuit eye movements. Smooth pursuit eye movements were included in the questionnaire as control questions, and we hypothesized that possible impairments in visual stability would selectively affect saccades. Participants were also asked if they had trouble estimating the positions of objects, had visual hallucinations, or had experienced any other visual symptoms. Answers were given on a four step scale (0 = never, 3 = often).

In the double saccade task the two targets are typically presented so briefly that when the participant initiates the first saccade, both visual targets have disappeared from the screen. Thus the participant must use of CD to take into account the spatial change produced by the first saccade in order to correctly fixate the second target (*Hallett & Lightstone, 1976*; *Sommer & Wurtz, 2008*; *Thakkar et al., 2015*). Pilot experiments for the present study showed that the traditional double saccade task was too challenging for the elderly participants (the participants had trouble perceiving and making saccades to the locations of the briefly presented stimuli). In order to make the task easier, we modified the task so that each target stimulus is presented for a longer duration, but the participants were instructed to initiate the saccades only after both targets had disappeared from the screen (Memory condition; Fig. 1B). While this means that short-term visuospatial memory may contribute to performance more in our modified version than in the traditional form of the task (which taxes the sensory memory more), the key idea behind the double saccade task remains unchanged. In the Baseline condition (Fig. 1A), which was physically identical to the experimental condition, the participants were not required to wait until the targets have disappeared and could initiate their eye movements voluntarily.

For the double saccade task, round white (45 cd/m$^2$) target stimuli with the dimension of 0.6° were presented on a black (three cd/m$^2$) background (i.e., Weber contrast = 46.6). The double saccade task began with a nine point calibration of the eye-tracker (i.e., whole screen was calibrated). After calibration, each trial began with a white fixation point presented at the center of the screen. This fixation point was also used as a (eye-tracking) drift-correction point in the beginning of each trial. After the participants correctly fixated the center of the screen for 1.2–1.5 s, the two target stimuli were presented. The first stimulus was always presented in one of the four quadrants of the screen (e.g., upper right quadrant as in Fig. 1A) for 500 ms ~16° from the fixation point. Right after the first target, a second target was presented for 130 ms on the horizontal meridian at the same side of the screen as the first target. The exact horizontal position of the second target varied from trial to trial (there were four equally likely positions, separated by a minimum of ~2.4°). Each participant performed the baseline and memory conditions once. A total of 30 trials were collected for Memory condition and 30 trials for Baseline condition (order counterbalanced). Before the experimental runs, each participant performed at least 10 practice trials without eye-tracking.

Visual 81% thresholds were measured by presenting a horizontally oriented, low-frequency, luminance modulated Gabor grating (diameter ~12°, 0.33 cycles per degree, phase varied randomly on each trial) in the upper or lower visual field (center of the Gabor was 7° from the center of the screen) for one screen refresh (11.8 ms). The stimuli were presented on mid-gray background. Participants' task was to report whether the Gabor stimulus was presented in the upper or lower visual field using the up and down arrow keys. The detection threshold was determined by adaptively varying the Michelson contrast of the Gabor using a Bayesian staircase method QUEST (*Watson & Pelli, 1983*). This procedure is similar to previously reported means of measuring saccadic suppression (*Burr, Morrone & Ross, 1994*). Each participant performed three different conditions (in separate blocks, order counterbalanced) during which visual thresholds were measured: fixation, saccade, and right after saccade. When the thresholds were measured during fixations, the participants kept fixating to a fixation dot, presented in the middle of the screen. In the "saccade condition," the visual threshold was determined for Gabor stimuli presented while participants performed ~19° saccades from a fixation point on the left side of the screen to a black dot at the center of the screen. In the "after saccade" condition the Gabor stimulus was presented ~100 ms after the saccade had ended. Participants always initiated the saccades voluntarily after a brief period of fixation during which the eye-tracker was drift corrected. To enable accurate tracking of horizontal eye-position, the eye-tracker was calibrated by three calibration points presented on the horizontal meridian (left side, right side, and center of the screen).

In both tasks (double saccade and visual threshold) the stimuli were presented using Psychtoolbox (*Brainard, 1997*) running on Matlab 2014b. Eye movements were recorded using EyeLink 1,000 eye-tracker (SR Research) that was operated using the EyeLink Toolbox (*Cornelissen, Peters & Palmer, 2002*). The eye movement registration was done monocularly, typically for the right eye, using 1,000 Hz sampling frequency. The stimuli were presented on a 19″ CRT screen with a screen resolution of 1,024 × 768 pixels and 85 Hz refresh rate. Participants were seated 70 cm from the screen and a head rest was used to stabilize the head.

## SPECT imaging

To image DAT binding, all PD patients underwent brain [$^{123}$I]FP-CIT ([$^{123}$I] *N*-ω-fluoropropyl-2β-carbomethoxy-3β-(4-iodophenyl) nortropane) SPECT. The interval between the scanning and behavioral testing (e.g., double saccade task) was 8–29 months. DAT binding a relatively stable parameter which has been reported to decrease on average 1–5% per year (*Kaasinen et al., 2015*; *Pirker et al., 2002*, *2003*). All of the reported results were also replicated when the models included a covariate denoting the delay between scanning and behavioral testing. Prior to scanning, thyroidal update of the ligand was blocked by administering oral potassium perchlorate (250 mg) 60 min before the injection of [$^{123}$I]FP-CIT. An 185 MBq bolus of [$^{123}$I]FP-CIT was administered and scanning performed 3 h after the injection. Imaging was performed as described earlier (*Kaasinen et al., 2014*). The images were reconstructed and analyzed using BRASS (Hermes Medical Solutions AB, Stockholm, Sweden). Specific binding ratios (SBRs) for the caudate nucleus and the putamen were quantified using occipital cortex as the

reference region $SBR_{ROI} = (ROI–OCC)/OCC$, where ROI refers to the uptake in the region of interest and OCC in the occipital cortex. Although occipital cortex may contain minute concentrations of DAT, it has been shown to be a valid reference region in $[^{123}I]$ FP-CIT imaging (*Joutsa, Johansson & Kaasinen, 2015*). Mean striatal values (caudate, anterior putamen, and posterior putamen) in the left and right hemisphere were used for correlation analyses. In statistical analysis of the relationship between DAT binding and behavioral performance, the analyses always included a factor that indicated whether saccades were made ipsilaterally or contralaterally with respect to the hemisphere with lower binding.

## Data analysis

Group level differences and effects of within-subject experimental manipulations on average saccade performance were analyzed using linear mixed-effects models using the lme4 package (*Bates et al., 2014*) in R statistical software (*R Development Core Team, 2014*). The analysis is performed on a single-trial basis rather than on aggregated means, which means that the mixed-effects models take into account within-participant variation in addition to group-level effects (*Baayen, Davidson & Bates, 2008*). Participants were included in the model as random variables, that is, each individual participant's data was fitted with individual intercepts and slopes (concerning the Baseline/Memory condition). The fixed-effect predictors included in the models are described in the Results section. The models were pruned by removing fixed-effects predictors that did not significantly ($t < |2|$) contribute to the model. The visual detection thresholds (saccadic suppression) were analyzed using linear mixed effects models with separate intercepts for each subject. As there is no standardized way to calculate the degrees of freedom and the corresponding $p$ values in mixed-effects models (*Baayen, Davidson & Bates, 2008*), we do not report $p$ values. However, the statistical significance of an effect can be checked from the confidence intervals (CI): whenever the value zero is included in the CI, the effect is not considered statistically significant. Another reason to not report $p$ values is to draw the reader's attention to the size of the effect (*Cumming, 2012*). Error bars in all the figures are 95% CIs calculated from 1,000 bootstrap samples.

In the double-saccade task, separate models were built for horizontal and vertical errors (relative to the target location). Random variation between participants concerning the intercept and the effect of the condition was included in the models. Only trials where the first saccade was directed toward the first target and where the participants made a following saccade toward the second target were included in the analysis (5.8% of trials were excluded). For each participant, the fixations closest to the first and second target locations were included in the analysis. The participants typically returned their gaze to the center of the screen in one large amplitude saccade after they had made the saccade toward the second target location. Before the analysis of fixation accuracy, the data was transposed to correspond to the case where the first target was at the upper right corner of the screen and the second target directly below it. These preprocessing steps were performed in Matlab 2014b.

Datasets can be downloaded from the Open Science Framework (https://osf.io/bqnzh).

## RESULTS

### Visual symptoms (questionnaire)

Parkinson's disease patients reported experiencing somewhat more general confusion about locations of objects, and experiencing spatial shifts during saccades compared to healthy controls, but this difference was not statistically significant (Mann–Whitney test; Table 1). Two patients (and none of the control participants) reported general confusion about object locations or reported noticing spatial shifts during saccades.

### Saccadic suppression

Visual thresholds did not differ between the control group and the PD group during fixation ($t = -0.13$), saccade ($t = 0.15$), or after saccade ($t = -0.91$). These factors were hence removed from the linear model used to analyze visual threshold. The pruned model showed that both groups displayed clear saccadic suppression as visual thresholds increased during saccades ($\beta = 1.96$, (95% CI [1.48–2.44], $t = 8.06$) and right after saccades ($\beta = 0.78$, CI [0.30–1.25], $t = 3.21$) when compared to thresholds during fixation (see Fig. 2). This indicates that the patients' visual sensitivity was similar to the control group during fixation, saccades, and right after saccades.

### Double-saccade task

#### Horizontal and vertical errors

Figure 3 shows the participants' accuracy in fixating the first and second targets. Note that for the saccades toward the first target negative values indicate hypometric saccades, whereas for the second target positive (vertical) values are a sign of hypometric saccades. Visual inspection of the data suggests that the performance of the two groups (PD vs. control) did not significantly differ from each other in the Baseline condition. However, in the Memory condition the PD patients' saccades to the first target were hypometric, and the saccades to the second target were biased toward the fixation when compared to the control participants. The data were analyzed using linear mixed-effects models that included the predictors group (Control vs. PD), condition (Baseline vs. Memory), target (T1 vs. T2), and their interactions. Horizontal and vertical errors were modelled separately. The results of the regression models are presented in Table 2 (horizontal error) and Table 3 (vertical error).

As shown on Fig. 3, fixations to the first target were somewhat hypometric, whereas fixations to the second target tended to overshoot the target slightly (as shown by the intercepts of the regression models). In both groups, fixations to the second target were vertically more accurate than fixations to the first target (Target: $t = 5.11$). When compared to the baseline condition (i.e., visually-guided saccades), memory-guided saccades were horizontally more accurate in both groups (Condition: $t = 2.07$). However, the PD patients' saccades to the first target were vertically more hypometric in the Memory condition (PD $\times$ Condition: $t = -3.16$). Moreover, whereas the fixations to the second target were biased horizontally outward in the memory condition in control participants (Target $\times$ Condition: $t = 2.19$), similar effect was not observed in the

| Table 1 Visual symptoms (mean and SD) | | | | | |
|---|---|---|---|---|---|
| Predictor | | Controls | PD | Z | p |
| General | Nausea | 0.07 (0.25) | 0.06 (0.24) | −0.05 | 0.96 |
| | Spatial confusion | 0.00 (0.00) | 0.13 (0.34) | −1.39 | 0.16 |
| | Hallucinations | 0.00 (0.00) | 0.00 (0.00) | 0.00 | 1.00 |
| Saccade | Vertigo | 0.14 (0.51) | 0.13 (0.33) | −0.45 | 0.65 |
| | Spatial shift | 0.00 (0.00) | 0.13 (0.33) | −1.39 | 0.16 |
| | Motion | 0.07 (0.25) | 0.13 (0.33) | −0.57 | 0.36 |
| Smooth pursuit | Vertigo | 0.00 (0.00) | 0.00 (0.00) | 0.00 | 1.00 |
| | Spatial shift | 0.00 (0.00) | 0.00 (0.00) | 0.00 | 1.00 |
| | Motion | 0.00 (0.00) | 0.00 (0.00) | 0.00 | 1.00 |
| Total | | 0.03 (0.09) | 0.07 (0.11) | −1.13 | 0.26 |

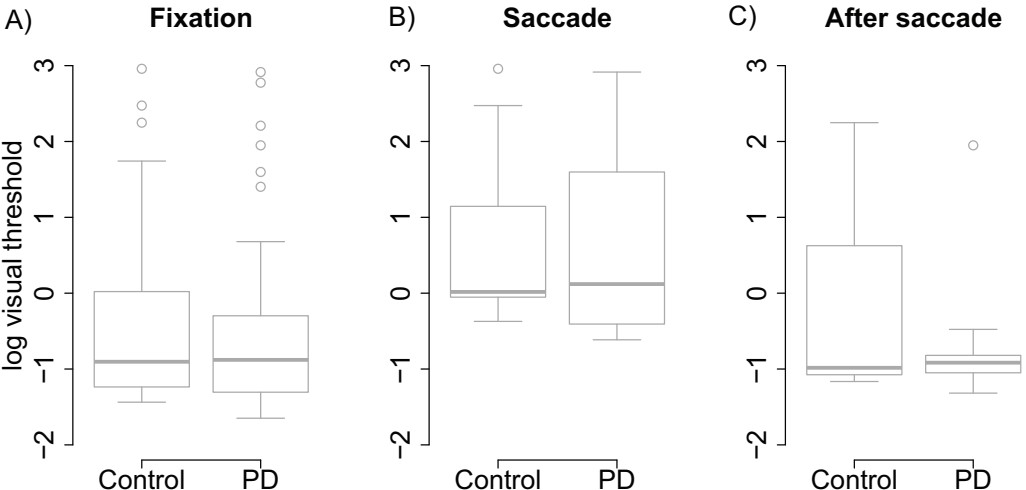

**Figure 2 Visual thresholds during fixation, saccade, and after saccade for each group.** (A) Fixation. (B) Saccade. (C) After saccade. The unit of the *y* axis is the logarithm of Michelson contrast of the stimulus. The bold horizontal line depicts the median and the box is the interquartile range. The ends of the whiskers are 1.5 times the interquartile range from the box. Circles are outliers.

PD patients (PD × Target × Condition: $t = -2.19$). In other words, consistent with amplified CD relative to motor commands, when making memory guided-saccades, the PD patients' fixations toward the second target were biased toward the center of the screen when compared to control participants. This effect could simply reflect the fact that the patients' saccades toward the first target were hypometric. For this reason, we also examined trial-by-trial variation in how well changes in the endpoint of the first saccade are taken into account when making the second saccade (see section: Compensation in angle and amplitude following the first saccade). Concerning errors in the vertical direction, the results showed that whereas control participants tended to overshoot the second target in the memory condition (Target × Condition: $t = -3.53$), such effect was not observed in patients (PD × Target × Condition: $t = 5.3$).

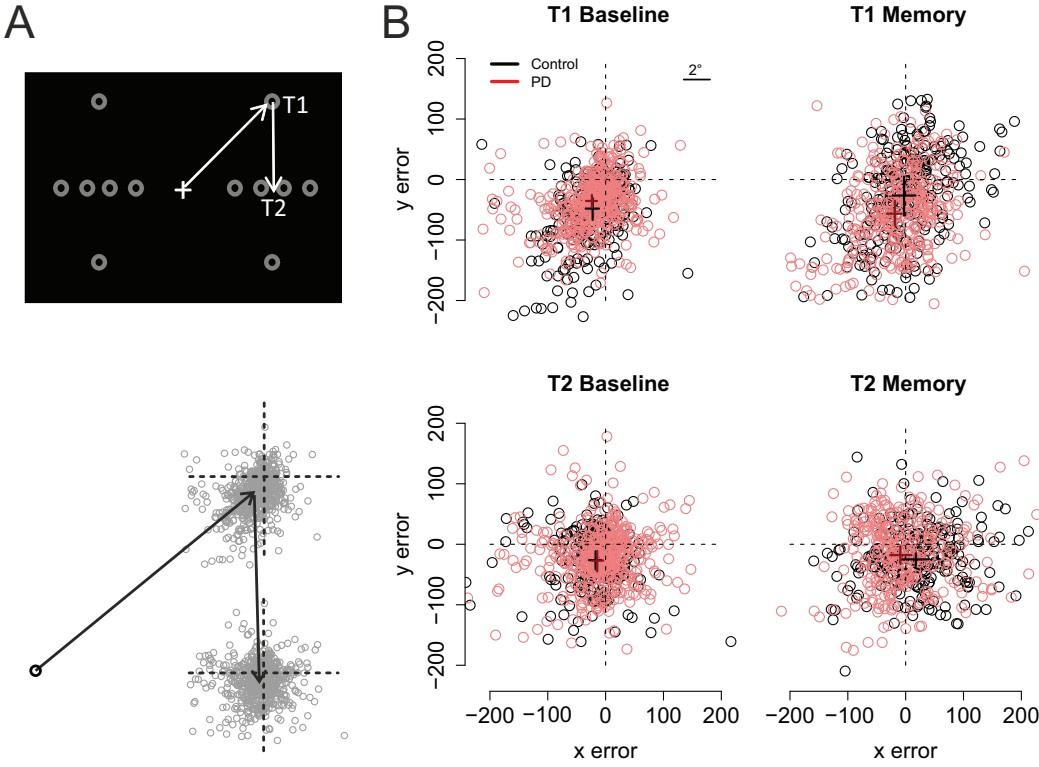

**Figure 3 Fixation maps.** (A) The grey circles show the possible positions of the targets. For data analysis, the coordinates of the targets/eye positions were transposed to correspond to the case where the first target was at the upper right corner of the screen and the second target directly below it. (B) Fixation maps for the first target (T1) and second target (T2), separately for Baseline condition and Memory condition. The black and red dots denote individual fixations of control participants and PD patients, respectively. The black and red crosses show the respective group averages; the lengths of the lines are 95% confidence intervals of the mean. Unit of error is pixels (22 pixels is ~1°, see inset).

**Table 2 Results of the linear mixed-effects model. Dependent variable: horizontal errors.**

| Predictor | Estimate | 95% CI | t value |
|---|---|---|---|
| Intercept | −22.99 | [−34.39, −11.55] | −3.92 |
| PD | −2.50 | [−17.69, 12.56] | −0.32 |
| Target (= T2) | 5.09 | [−4.34, 14.47] | 1.05 |
| Condition (= memory) | 20.85 | [1.25, 39.35] | 2.07 |
| PD × Target | 8.29 | [−3.36, 19.92] | 1.39 |
| PD × Condition | −12.95 | [−38.40, 12.30] | −0.99 |
| Target × Condition | 14.85 | [1.57, 28.08] | 2.19 |
| PD × Target × Condition | −18.96 | [−35.89, −2.00] | −2.19 |

As motor symptoms in PD are often asymmetric (see *Choi et al., 2011*), for an example concerning eye movements), we analyzed if the accuracy of fixations was modulated by whether the targets were presented ipsi- or contralaterally with respect to the predominant motor symptoms. This analysis was restricted to PD patients who showed asymmetric

**Table 3 Results of the linear mixed-effects model. Dependent variable: vertical errors.**

| Predictor | Estimate | 95% CI | t value |
|---|---|---|---|
| Intercept | −49.48 | [−64.09, −34.83] | −6.59 |
| PD | 11.48 | [−8.00, 30.90] | 1.15 |
| Target (= T2) | 23.74 | [14.70, 32.93] | 5.11 |
| Condition (= memory) | 23.94 | [5.04, 42.87] | 2.46 |
| PD × Target | −8.72 | [−20.03, −2.52] | −1.51 |
| PD × Condition | −40.86 | [−66.08, −15.71] | −3.16 |
| Target × Condition | −23.18 | [−36.04, −10.37] | −3.53 |
| PD × Target × Condition | 44.45 | [28.06, 60.87] | 5.3 |

symptoms ($N = 12$). The model included factors condition (Baseline vs. Memory), target (T1 vs. T2), a laterality factor (indicating whether the targets were presented ipsi- or contralaterally with respect to primary motor symptoms), and their interactions. The data are presented in Fig. 4. The analysis showed that horizontal errors increased (biased toward to the center of the screen) when saccades were made toward the side with primary motor symptoms (main effect of target laterality: $\beta = -11.18$, CI [−18.30, −4.06], $t = -3.07$), and that this effect was stronger in the Memory condition (laterality × condition: $\beta = -14.77$, CI [−25.82, −3.72], $t = -2.61$). Target laterality (with respect to primary motor symptoms) did not modulate vertical errors.

*Modulation by DAT binding.* Finally, we wanted to see if the accuracy of fixations correlated with striatal DAT binding in PD patients. This analysis focused on mean striatal DAT binding (caudate, anterior putamen, and posterior putamen). In addition to the condition (Baseline vs. Memory), the model included predictors condition, target, the overall DAT binding in the more affected hemisphere (i.e., the one with lower binding), the motor MDS-UPDRS score of the patient, and the interactions of these predictors. We observed that DAT binding modulated vertical errors ($\beta = 25.40$, CI [3.74–46.89], $t = 2.40$): the larger the DAT binding, the more accurate were the fixations. This result is visualized in Fig. 5A. Note that Fig. 5A displays fixations to both targets because the model suggested that Target did not modulate the correlation. No effects for DAT binding were observed for horizontal errors.

We hypothesized that the effect may be better detected if we analyze the amplitudes of the saccades (not the accuracy of fixations). The results showed that DAT binding weakly (not statistically significantly) correlated with saccade amplitudes (main effect of DAT binding: $\beta = 27.69$, CI [−1.75–57.16], $t = 1.8$), and a stronger effect was observed in the Memory condition for saccades to the second target (DAT × Condition × Target: $\beta = 42.38$, CI [14.96–69.80], $t = 3.02$). These effects are visualized in Fig. 5B (saccades to first target) and Fig. 5C (saccades to second target).

Importantly, all correlations with DAT binding were also observed when the MDS-UPDRS motor score of the participant was included in the model. This suggests that the observed correlations represent true correlation with the integrity of the dopaminergic system, and not just the severity of the PD.

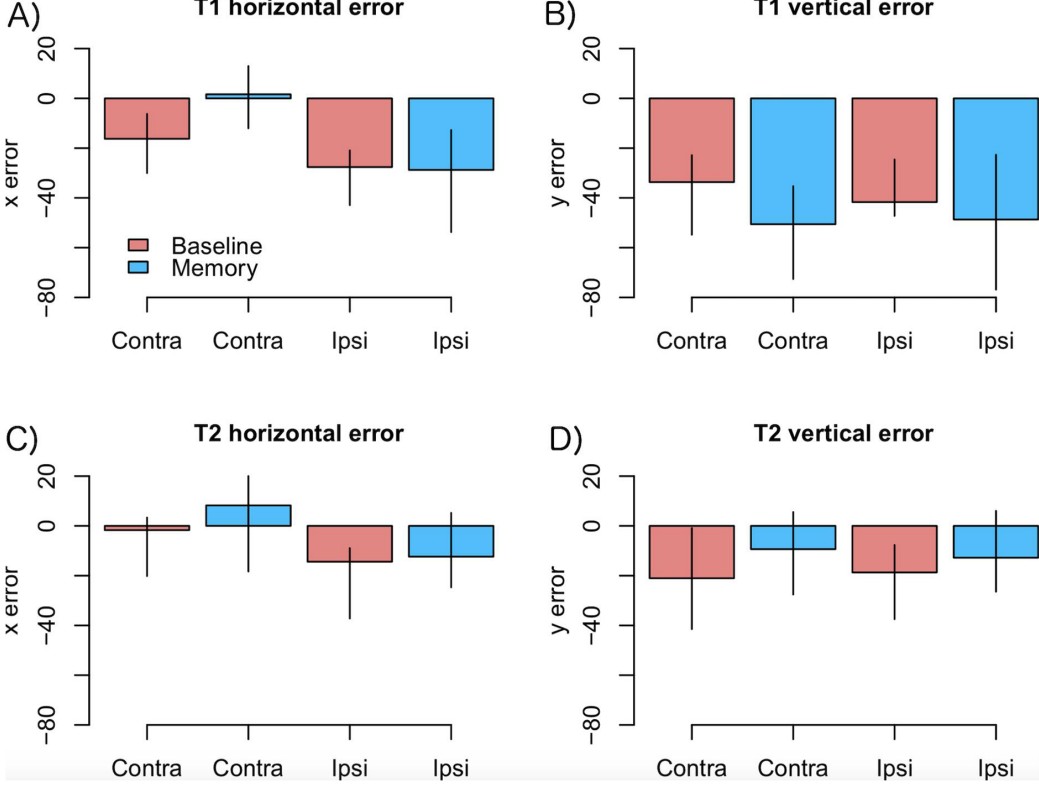

**Figure 4 Amount of error as a function of laterality in the PD group.** (A) T1 horizontal error, (B) T1 vertical error, (C) T2 horizontal error, (D) T2 vertical error. Observed data, not model results. Ipsilateral refers to targets that were presented at the same side as the motor symptoms. Error bars are the 95% CIs of means. Unit of error is pixels (22 pixels is ~1°).

### Compensation in angle and amplitude following the first saccade

As explained in Fig. 1C we also examined how well the participants took into account the trial-by-trial variability in the first saccade when executing the second saccade. In the regression models, the predictor variables were the centered optimal angle/amplitude of the saccade (see Fig. 1C), group, condition, and all interaction between these three variables. Models were pruned by removing non-significant regressors.

As shown in Figs. 6A and 6B, the trial-to-trial variability in the *angle* of the second saccade was strongly predicted by the optimal angle required to perfectly fixate the second target ($\beta = 0.71$, CI [0.67–0.76], $t = 29.62$). In other words, the participants clearly had information about the error in the endpoint of the first saccade which they could use to correctly aim to the second target. In the Memory condition, the angle of the second saccade was smaller than in the Baseline condition. That is, participants' saccades (in both groups) were biased toward the center of the screen (i.e., the initial fixation point; $\beta = -4.69$, CI [−7.75, −1.67], $t = -3.09$). Furthermore, in the Memory condition, the larger the optimal angle, the larger was the bias toward the center of the screen (interaction: $\beta = -0.13$, CI [−0.20, −0.02], $t = -4.32$). PD patients were not statistically significantly different from control participants ($t \leq |1|$).

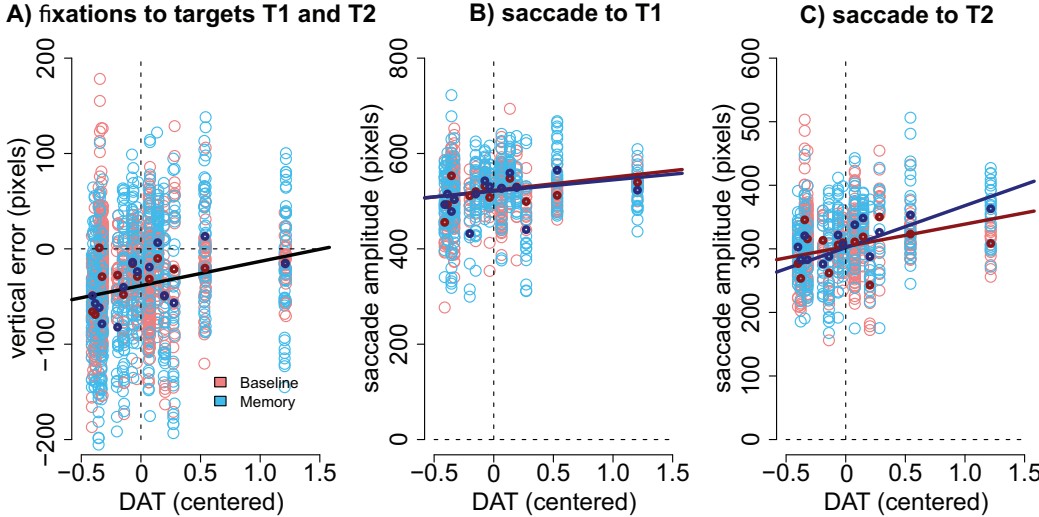

**Figure 5 Correlation between DAT binding and saccades.** (A) Vertical accuracy of fixations was modulated by DAT binding. The plot shows fixations to both the first and the second targets. (B) The amplitudes of saccades to the first target did not statistically significantly correlate with DAT binding in either condition. (C) The amplitudes of the saccades toward the second target correlated with DAT binding in the Memory condition. DAT binding variable has been centered (i.e., zero represented the mean). The lighter dots represent the observed data (all observations), and the bold dots represent the participant-wise averages of these data. Lines represent the observed correlation with DAT binding from the mixed-effects models. Red color denotes the baseline condition (visually-guided saccades) and blue color the memory-guided saccade condition.

Also the *amplitude* of second saccade strongly correlated with the optimal amplitude ($\beta = 0.87$, CI [0.78–0.96], $t = 19.46$). Moreover, although the amplitudes of the second saccades did not differ between groups ($t = 0.04$), there was an interaction between group and optimal amplitude: Patients' second saccades become more hypometric as the optimal amplitude required to reach the target increased (Figs. 6C and 6D, blue line: $\beta = -0.15$, CI [−0.27, −0.04], $t = 2.61$). In other words, the patients had trouble making long saccades.

As with previous analyses, we examined whether the laterality of the motor symptoms correlated with the angles/amplitudes of the second saccades. As shown in Fig. 7, when the targets were presented at the same side as the motor symptoms the compensation in saccade angle was worse than when the targets were presented on the side with less motor symptoms—the second saccades become biased toward the center of the screen, consistent with amplified CD ($\beta = -0.11$, CI [−0.19, −0.033], $t = -2.78$). In other words, although the PD group did not in general as a group differ from healthy controls in compensating for variability in the endpoint of the first saccade, such an effect was observed when the laterality of the symptoms was considered. In other words, this means that since in some patients the symptoms are predominantly on the right side, and in other patients on the left side, the deficit is only observed when the laterality of the symptoms is taken into account.

Target side relative to motor symptoms did not modulate the precision of compensation for saccade *amplitudes.*

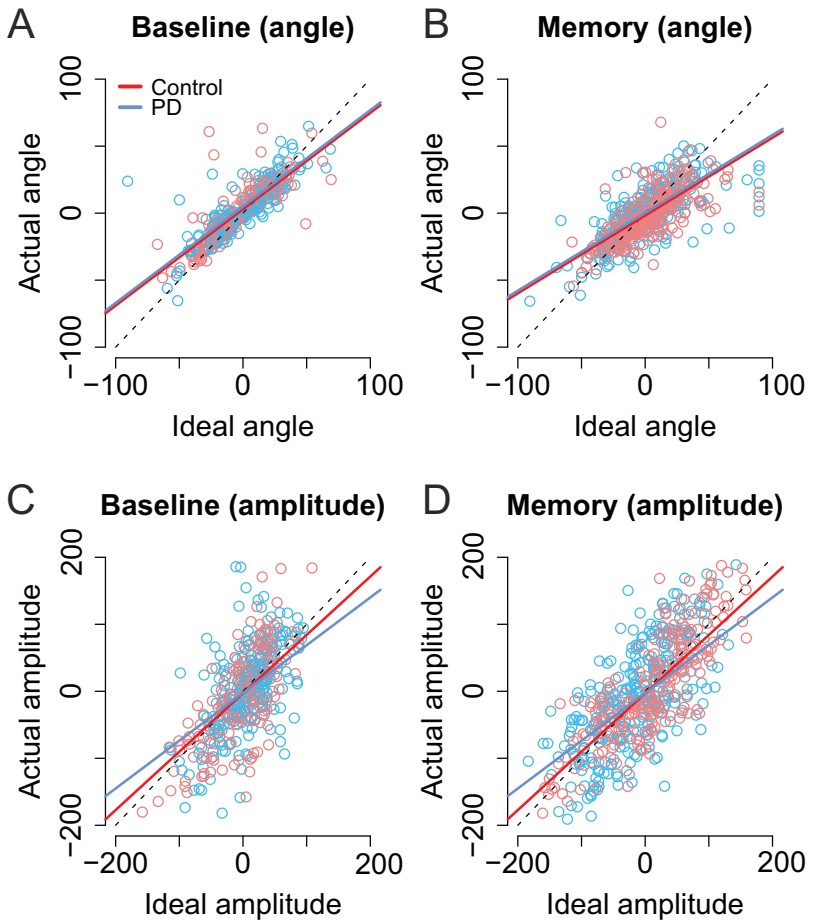

**Figure 6 Second saccade angles (A–B) and amplitudes (C–D) as a function of ideal angle/amplitude.**
Left panels show the baseline condition, and the right panel the Memory condition. Red color denotes
the healthy control participants, and the blue color the patient group. Dots show the observed data
(individual saccades) and lines the results of the fitted linear mixed-effects models. Note that angle/
amplitude variables are centered, that is, negative values show that angle/amplitude was smaller than
average. Unit is pixels (22 pixels is ~1°).

*Modulation by DAT binding.* We also examined whether the success of compensating for
variability in the endpoint of the first fixation correlated with DAT binding in PD patients.
As previously, the model included condition, target laterality (with respect to the more
affected hemisphere), the amount of DAT binding, the optimal angle to perfectly fixate the
second target, and the motor MDS-UPDRS score of the participant (and the interactions
between these regressors).

As previously stated the actual *angles* of the saccades were strongly predicted by the
optimal angle required to reach the second target ($\beta = 0.74$, CI [0.67–0.81], $t = 20.3$).
As shown in Fig. 8A (blue dashed line) patients with lower DAT transporter binding
values were more inefficient in taking into account for the trial-to-trial variability in the
endpoint of the first saccade and their saccades become more biased toward the center
of the screen, consistent with stronger reliance on CD than bottom-up visual information
(Ideal angle × DAT: $\beta = 0.17$, CI [0.070–0.28], $t = 3.2$). Furthermore, the modulation

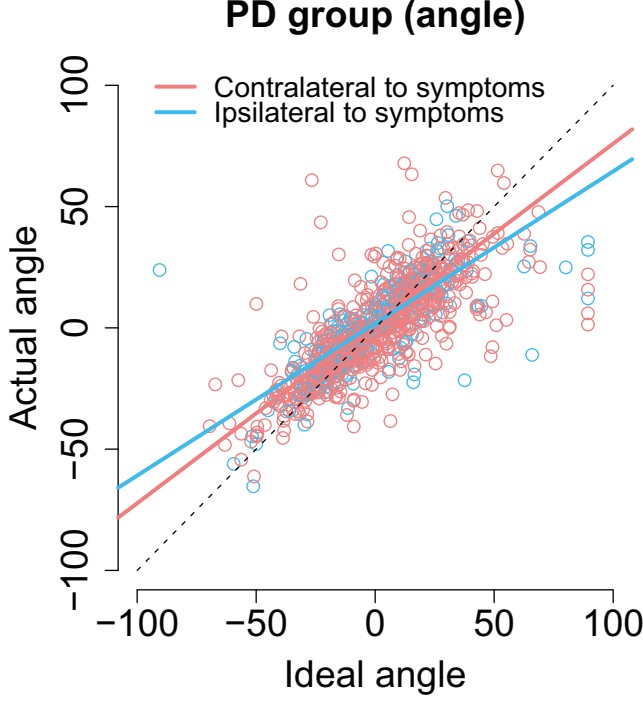

**Figure 7 The angle of the second saccade as a function of ideal angle.** The solid lines show the results of the model. Compensation for variation in the endpoint of the first fixation is worse for targets that are at the predominant side of motor symptoms. Compensation for variation in the endpoint of the first fixation is worse for targets that are at the predominant side of motor symptoms.

of the angle of the second saccade by DAT was not explained by the severity of PD as the motor MDS-UPDRS score revealed similar modulatory effect (Ideal angle × UPDRS: $\beta = 0.0030$, CI [0.00081–0.0053], $t = 2.6$). Note that although the $\beta$ coefficient of the Ideal angle × UPDRS interaction (0.0030) is much smaller than the $\beta$ coefficient of the Ideal angle × DAT interaction (0.17), the relative size of this effect is in fact very similar (due to the fact that the range of motor MDS-UPDRS scores is much higher than DAT values). The modulatory effect by motor MDS-UPDRS is visualized in Fig. 8A as the red dotted line which overlaps the blue dashed line that depicts modulation by DAT. In addition to the above mentioned two-way interactions, a Ideal angle × motor MDS-UPDRS × DAT interaction was observed ($\beta = 0.014$, CI [0.004–0.025], $t = 2.8$). This three-way interaction indicates that the modulation by DAT binding was stronger in participants with more severe PD (i.e., higher UPDRS scores).

Next, we examined whether DAT binding was associated with the ability to modulate the *amplitude* of the second saccade to compensate for trial-to-trial variability in the first saccade. Although patients were in general successful in making a second saccade whose amplitude closely resembled the ideal amplitude required to perfectly fixate the second target ($\beta = 0.73$, CI [0.65–0.81], $t = 17.7$), patients with lower DAT binding made more hypometric saccades ($\beta = 53.94$, CI [19.23–88.65], $t = 3.0$), as shown in Fig. 7B. UPDRS score did not statistically significantly modulate saccade amplitudes in general

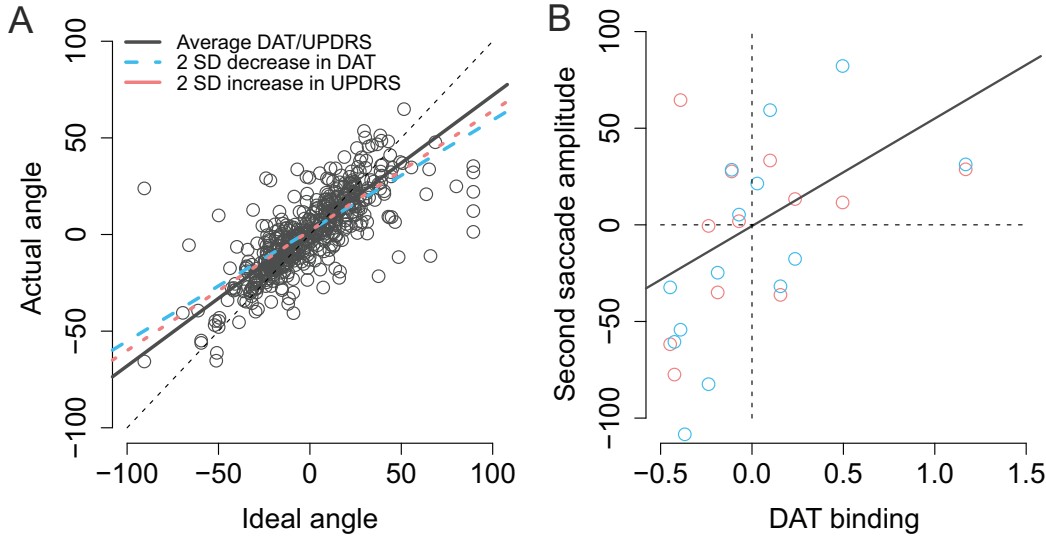

**Figure 8 Modulation of compensation in the amplitude of the second saccade.** (A) Compensation for variation in the endpoint of the first fixation is modulated by DAT binding (blue dashed line) and MDS-UPDRS (red dotted line). The visualized effects represent two SD decrease (DAT) or increase (MDS-UPDRS) relative to the patient group average, which is plotted as a gray solid line. The circles represent the observed data. (B) Correlation between the amplitude of the second saccade (centered) and DAT binding (centered). The red and blue circles represent the (average) observed data in the Baseline and Memory condition respectively. The gray line shows the modelled results (Baseline and Memory conditions pooled, as no Condition × DAT interaction was observed).

(main effect of UPDRS: $t = -0.3$). However, patients with higher motor MDS-UPDRS scores were more inefficient in making a close-to-ideal amplitude saccade to the second target (Ideal amplitude × motor MDS-UPDRS: $\beta = -0.010$, CI [$-0.015$, $-0.0050$], $t = -3.7$). Surprisingly, this effect was only observed when the targets were presented ipsilaterally with respect to the hemisphere with lower DAT binding (Contralateral × Ideal amplitude × motor MDS-UPDRS: $\beta = 0.012$, CI [$0.0048$–$0.20$], $t = 3.1$).

## DISCUSSION

We examined how well early PD patients can update their spatial representations during eye movements, and whether this ability correlates with the integrity of the dopaminergic system. The participants were asked to make either visually-guided or memory-guided eye movements to two targets presented in succession. First, the observed results replicate the previous finding that PD patients have difficulties in making memory-guided saccades when compared to control participants (*Kimmig et al., 2002*; *Blekher et al., 2009*; *Crawford, Henderson & Kennard, 1989*; *Terao et al., 2011*). Second, we observed that the deficits in saccadic eye movements correlated with striatal (caudate, anterior putamen, and posterior putamen) DAT binding. Third, our results suggest that in addition to general impairments in making saccadic eye movements, the patients have deficits in internally monitoring their eye movements: The angle of the saccade toward the second target was biased in a manner that suggests that the patients overestimated the length of their first saccade. That is, the errors in the angle of the second saccade cannot be

explained by simply assuming that PD produces hypometric movement. We suggest that these errors reflect an impairment in monitoring their own movements. As discussed below, the which exact mechanisms explains this deficit, remains open.

## Internal monitoring of movements

When examining the accuracy of fixations, PD patients' fixations toward the second target were on average biased toward the center of the screen. This effect is consistent with the hypothesis that PD patients' CD signals indicate that a sufficiently large eye movement was made, when in reality the eye movement was hypometric. However, the effect could reflect the fact that the patients' saccades toward the first target were hypometric. On the other hand, one could argue that the result shows that the patients did not notice that their first saccade was hypometric, and hence were not able to correct for the saccadic hypometria when performing the second saccade. To tease apart the monitoring of eye movements from saccadic hypometria, we also examined whether the patients' ability to correct for the variation in the endpoint of the first saccade was impaired when compared to the control participants. The results showed that in addition to being hypometric, the angle of the second saccade was biased in a manner consistent with amplified CD relative to the actual movement in PD patients. The effect was only observed when the patients made saccades toward the side with predominant motor symptoms. The bias in the angle of the second saccade (in PD patients) cannot be explained simply by saccadic hypometria. However, the effect was observed both during visually-guided and memory-guided saccades (Fig. 8A), which is at odds with our hypothesis that deficits in CD are only observed in the Memory condition. The result may indicate that the deficit is not in the CD, but instead reflects some more general impairment in encoding or updating spatial representations after eye movements.

The observed findings are consistent with the mechanism proposed by *Friston et al. (2012b)*: dopaminergic deficiency leads to reduced precision in bottom-up sensory information (prediction errors, such as visual input or proprioceptive input from extraocular muscles), which causes the brain to rely more on top-down information (i.e., predictions such as motor programs and commands). This means that the hypokinetic movements (predictions) begin to dominate action, because feedback concerning the performed, hypometric movements (prediction errors) is not sufficiently assimilated into the current model of the world. This could lead to the fragmented, multi-step movement patterns observed in PD (*Berardelli et al., 2001*; *Kimmig et al., 2002*), especially in situations where visual information is not available to guide behavior (*Glicksterin & Stein, 1991*; *Klockgether et al., 1995*; *Jobst et al., 1997*). Such mechanism could also result in an impaired ability to take into account visual information (prediction errors) when performing actions: Due to the reduced precision in incoming visual information (i.e., location of the second target), the movements toward the target will be predominated by the internal model of the patient.

Previous research has shown that deficits in CD are associated with *reduced* CD relative to motor signals (*Sommer & Wurtz, 2002*; *Thakkar et al., 2015*), that is, converse to what we observed here. However, there is no reason to assume that deficits in CD

always results in reduction of the CD signal (although this is obviously the case when an area that conveys the CD signal is lesioned; *Sommer & Wurtz, 2008*). The present findings are consistent with the finding that schizophrenic patients (who in contrast to PD patients have hyperactive striatal dopamine neurotransmission (*Howes & Kapur, 2009*)) reveal the reverse pattern of results when performing the double-saccade task: in contrast to the present results, schizophrenic patients seem to underestimate the size of the first saccade in the double-saccade task (*Thakkar et al., 2015*). This conclusion rests on the assumption that the dopaminergic system contributes to the monitoring of movements. Future studies should aim to directly compare these two groups of patients in the same study.

## Contributions of the dopaminergic system to saccades

The integrity of the striatal dopaminergic system was observed to correlate with the accuracy of saccadic eye movements. The smaller the DAT binding, the more hypometric were the patients' saccades (Fig. 5B), as previously reported (*Hotson, Langston & Langston, 1986*; *Brooks, Fuchs & Finocchio, 1986*; *Schultz et al., 1989*; *Kato et al., 1995*). Second, our results suggest that in addition to correlating with saccadic hypometria, the integrity of the dopaminergic system may also correlate with the ability to monitor one's own eye movements. This conclusion is based on the finding that the angle of the second saccade was more optimal in patients with higher DAT binding (Fig. 8A), and that DAT binding more strongly correlated with amplitudes of saccades to the second target in the Memory condition (Fig. 5). These findings may, however, also reflect dopaminergic contributions to maintaining information in memory (*Landau et al., 2009*; *Costa et al., 2003*). Altogether the observed correlation between DAT binding and saccadic metrics (hypometria and the angle of the second saccade) are consistent with the proposal that the dopaminergic system contributes to taking into account previous movements and contextual information when making sequential movements (*Friston et al., 2012b*).

## Limitations of the present study

The present study has important limitations. First, the sample size of our study was small and possible confounding effects of medication cannot be ruled out. However, because out of 14 patients only two chose to take their morning medication, we consider unlikely that acute medication effects were involved in the described effect. The present study's power to detect correlations between DAT binding and saccade performance is very limited, and the small sample size may also lead to inflated correlations (*Yarkoni, 2009*). Second, although our results suggest that the patients' internal monitoring mechanisms signaled a larger movement than was actually performed, the size of this result is relatively small (e.g., compared to the size of saccadic hypometria for saccades toward the first target). This could in part be explained by the fact that the patients learned the locations of the targets and could use, for example, the monitor's frame to guide their eye movements. Similarly, a better approach would have been to employ novel, unpredictable target locations on every trial (*Hodgson et al., 1999*). The present study employed a modified version of the double-saccade task. In the classical version

(*Hallett & Lightstone, 1976*; *Sommer & Wurtz, 2002*; *Thakkar et al., 2015*) the visual targets are presented so briefly that they are removed from the screen before the eye movements begin; hence the saccades cannot be guided by sensory information. Because the classical double-saccade task-set up proved too demanding for elderly subjects, we used a modified task where the targets were presented for a longer duration. Third, we cannot rule out the possibility that the observed spatial bias in the Memory condition in PD patients results from impaired spatial memory, rather than deficits in internal monitoring of movements, although impaired spatial memory does not in itself imply that saccades are biased toward the center of the screen. Fourth, it is possible that the observed correlations between fixation errors and dopamine reflect, for an example, *combining* motor commands to sequences (not integrating information about what type of movement was made). This interpretation may in part explain correlations between the *amplitude* of the second saccade and DAT binding, but it cannot explain why the *angle* of the second saccade is spatially biased toward the initial fixation. Finally, the present conclusions are restricted by the correlative nature of our study. Causal evidence could be acquired by experimentally manipulating dopaminergic activity in healthy humans.

## CONCLUSION

Our results provide preliminary evidence that the patients may have impairments in incorporating information about previously made movements and incoming visual information into their current model of the world. The observed disturbances in eye movements, together with the shown associations with DAT binding suggest that, as the disease progresses, and the dopaminergic deficit deepens, relevant clinical motor and visual symptoms associated with impaired internal monitoring of movements may emerge. The PD patients in the present study were at early stages of PD, did not report visual symptoms, and the visual sensitivity of the patients was similar to the control group. Advanced PD patients have many visual symptoms (*Weil et al., 2016*) and impaired internal monitoring of movements could lead to deficits in experienced visual stability in later stages of PD.

### Funding

Henry Railo was funded by Turku Institute of Advanced Studies and Academy of Finland (grant #308533), and received a grant from the Finnish Cultural Foundation. Henri Olkoniemi was funded by the Finnish Cultural Foundation. Juho Joutsa was funded by the Academy of Finland (grant # 295580), the Finnish Medical Foundation and the Orion Research Foundation, and has received travel grants from Orion and Abbvie, and a research grant from Lundbeck. Valtteri Kaasinen was funded by the Turku University Hospital (ERVA-funds). The funders had no role in study design, data collection and analysis, decision to publish, or preparation of the manuscript.

## Grant Disclosures

The following grant information was disclosed by the authors:
Turku Institute of Advanced Studies and Academy of Finland: 308533.
Finnish Cultural Foundation.
Academy of Finland: 295580.
Finnish Medical Foundation and the Orion Research Foundation.
Orion and Abbvie, and a research grant from Lundbeck.
Turku University Hospital (ERVA-funds).

## Competing Interests

The authors declare that they have no competing interests.

## Author Contributions

- Henry Railo conceived and designed the experiments, analyzed the data, contributed reagents/materials/analysis tools, prepared figures and/or tables, authored or reviewed drafts of the paper, approved the final draft.
- Henri Olkoniemi performed the experiments, analyzed the data, contributed reagents/materials/analysis tools, authored or reviewed drafts of the paper, approved the final draft.
- Enni Eeronheimo performed the experiments, approved the final draft.
- Oona Pääkkönen performed the experiments, approved the final draft.
- Juho Joutsa analyzed the data, authored or reviewed drafts of the paper, approved the final draft.
- Valtteri Kaasinen conceived and designed the experiments, authored or reviewed drafts of the paper, approved the final draft.

## Human Ethics

The following information was supplied relating to ethical approvals (i.e., approving body and any reference numbers):

The experiment was approved by the Ethics committee of the hospital district of South-West Finland (ETMK 18/1801/2016).

## Data Availability

Railo, H., & Olkoniemi, H. (August 14, 2017). Double-saccade task / Saccadic suppression in Parkinson's disease (with DAT). Retrieved from https://osf.io/bqnzh/.

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
