# Peer review of "Dopamine and eye movement control in Parkinson’s disease: deficits in corollary discharge signals?"

_PeerJ, doi:10.7717/peerj.6038_

## Round 0.1 · original submission · Major Revisions

The authors have the opportunity to review the Manuscript which requires important work before being further considered. Please take in consideration in particular Reviewer 1 comments. They are pertinent, come from an expert in the field, and raise very important possible sources of weakness for the Manuscript.

Reviewer 1 ·

Basic reporting

This paper examines how patients at an early stage of Parkinson disease can update their spatial representations during eye movements, and whether this ability correlates with the integrity of the dopaminergic system. Although the topic is interesting, the paper has important limitation.

Experimental design

My main concern is about the analysis model. I am not convinced that this is the best model to apply to these data. In many cases, the data from patients and controls are very similar. I do not understand how the differences come from.

Validity of the findings

Another major issue is about the sample and the use of medications. The sample is small and very heterogeneous, subjects were on different medications and one person did not take any medication. This is an important limitation of the study because the different drugs may have different effects in the eye movements control.

Additional comments

Comments (in order of reading):
The introduction is exhaustive, however it is too long. I would suggest to the Authors to shorten it focusing on main issues, especially section 1.2.
Line 121: “CD signals…” please define before using abbreviations.
Line 186-187: this is an unusual way to indicate the sample. Please write “8 women, 6 men and 7 women, 7 men”. Also, please provide age information for both sexes separately.
Line 203: medication. 11 subjects did not take medications on the morning of the test, 2 patients took medications and one patients did not have medication prescription. How can the Authors be sure that interrupting the medication on the same day of recording is sufficient for avoiding a pharmacological effect? In addition, some subject did not interrupt medication, so there could be mixed effects in the results. Also, patients were on different medications. Again, some effect could be due to the specific drug.
Line 217: “We assumed that…” Did the Authors mean, “hypothesized”?
Line 245: please write “round white target stimuli with the dimension of 0.6⁰ were presented on a black background. Please specify the target intensity.
Line 246: Please specify the type of eye tracker and the sampling rate of recordings.
Line 252: 16 degrees is a big distance for making an accurate saccade. Did the Authors consider this aspect?
Section 2.2 needs to be shorten and rewritten. There are quite a few repetitions. For example, there are two paragraphs explaining the double saccade task used in this and previous studies (see lines 227 and 254). These two sections can be merged for a better reading.
Line 260: were the trials randomly presented?
Line 293: the time between the scanning and the test is very long and variable, a lot of changes could happen between 8 and 29 months. Did the Authors consider this aspect?
Line 324: as I said before, I am not convinced that this is the best model to analyze these data. Indeed, this model has no standardize way to calculate the significance level. The use of the confidence interval could result difficult for general readers.
Line 346: Authors should write a detailed legend of table 1. It is not clear to me which data the Authors are reporting here, is it a questionnaire? Is it a recorded data? How was it measured? Please describe.
Line 360: please write a detailed legend for figure 2. What does the horizontal line mean? What does the dots mean? Which data are represented by the bar?
Line 368: from the data presented in figure 3 it looks like patients and controls have a similar oculomotor behavior in response to the two task. Tables 2 and 3 show the errors made by PD patients, the authors should show the errors made by controls to demonstrate that the two groups differ.
Line 400: data from control participant are not shown in tables 2 and 3, or at least this is not explained anywhere. I strongly suggest the Authors to pay much more attention at the figures/tables explanation.
Line 426: which unit is represented in figure 4 x axes?
Line 441: “This suggests that the observed correlations represent true correlation with the integrity of the dopaminergic system…”. It is difficult to believe this statement because the subjects were on medications; furthermore, different medications. See also my previous comments.
Line 475-480: the Authors state that there is not a difference between PD patients and controls, however when “side” is taken into account the analysis model shows differences. I think that this is not the best way to analyze the data. The “side” should be inserted in the model as a factor.
Line 488-491: here the Authors state that the laterality is included in the model. This is very confusing (see previous comment). I suggest the Authors to pay much more attention to the description of the model.
Line 500: The Authors state that the β coefficient of the Ideal angle × UPDRS interaction is much smaller than the the β coefficient of the Ideal angle × DAT interaction, but the relative size of this effect is very similar due to the fact that the range of motor MDS-UPDRS scores is much higher than DAT values. The analysis should not work like that. Your model should tell you if there is an effect or if there is no effect, you cannot interpret the results according to the values of the data, this is wrong. If the scores are much higher than the DAT values and you do not rely the data to the result, your model could be wrong.
Line 507 and figure 8: see my previous comment; I am not convinced that this is correct.
Figures 7 and 8: what does the dashed lines mean? Is it the fitted effects from the model? Please describe.
At line 537 the Authors say “our results suggest that in addition to general impairments in making saccadic eye movements, the patients may also have deficits in monitoring their eye movements”, while at line 556 they say “The result may indicate that the deficit is not in the internal monitoring of movements”. To avoid misinterpretation, I would suggest to rewrite one of the two sentence or to better explain what the Authors mean with “internal”.
Line 574: See my previous comment about “internal”. It is better to avoid the use of “internal”, please be more specific. See also line 537 and 582.
Line 583: “…it also remains possible that the contrasting findings in schizophrenic patients and PD patients reflect different mechanisms”, this is obvious, please delete it or be more specific.

Reviewer 2 ·

Basic reporting

This study examined the ability to monitor eye movements possibly through corollary discharge signals of patients with Parkinson’s disease. Patients and healthy control participants performed a double-saccade task in which subjects were required to saccade to two briefly presented target sequentially. In this task the second saccade should be made taking into account the information about the errors of the first saccade. The same task was also performed with visual guidance.
Two main results are described.
First, the patients had difficulties to perform the second saccade taking into account the error in the generation of the first saccade when the second saccade was performed towards the side that was dominant for the motor symptoms. As the authors discussed the bias in the errors of the second saccade in the Parkinson’s patients were observed in both memory and visually guided movement ruling out that the initial hypothesis that the saccade errors depended only on a deficit of internal monitoring of eye movement. To account for the errors in both tasks the authors proposed a more complicated explanation that considered the possibility of an additional problem of precision of the bottom-up information.
Similarly, to previous studies it showed that a reduction in DAT binding was associate to more hypometric saccades, however it shows also a new result that the dopaminergic system can be important for eye monitoring because of the correlation between DAT binding and the amplitude of the saccade to the second target in the memory condition. However, spatial memory can also have a role for the longer duration of the stimulus presentation as discussed by the authors and the screen can be a reference frame to memorize the targets making harder to interpret the results.
Although the analyses are appropriate, the results are difficult to interpret for the similarity of the results in the visual and memory tasks. The title reflects the interpretation issue containing a question.

Experimental design

Although the analyses are appropriate, the results are difficult to interpret for the similarity of the results in the visual and memory tasks. The title reflects some of the interpretation problems being formulated as a question.

Validity of the findings

The interpretation of the findings should be explained better. The discussion should be improved because the interpretation of part of the results is unclear.
Line 558, “For instance, the patients may fail to assimilate visual information and information about the earlier saccade with their internal model of the world.” It is not clear this part. Does it mean that the bias in the angle of the saccade in the visually guided task (that is similar to the bias in the memory task) might be explained by a difficulty to assimilate visual information, while in the memory task the problem concerns eye movement monitoring? It is not clear to me.

Line 636, “Our results provide preliminary evidence that the patients may have impairments in incorporating information about previously made movements and incoming visual information into their current, internal model of the world.” I see a problem with this statement related to my previous question. This interpretation is true only in the visually guided task but not in the memory task. Please explain better.

Minor comments
The part in the introduction on prefrontal cortex should be extended to include more references to previous studies that have shown monitoring signals about previous actions such as an inactivation study of Postle and Tsujimoto (2012). For a review, consider on the effect of previous movements see Genovesio and Ferraina 2014.
Other papers on the influence of previous actions that might be considered. For example, the paper in the rats’ striatum of Kim H, Lee D, and Jung MW (2013), Signals for previous goal choice persist in the dorsomedial, but not dorsolateral striatum of rats.
Parietal cortex might also important for monitoring previous movements, see the paper of Genovesio, Brunamonti, Giusti, Ferraina (2007) on the influence of eye movement vectors and eye position.
I suggest changing the introduction by eliminating the subdivision in subparagraphs; it does not look to follow the journal style for the introduction.
Line 439, Use upper case

---

## Round 0.2 · accepted · Accept

Dear Authors, we thank you for the work done during the revision. The overall quality has improved substantially. Congratulations.

# Reviewer 1 ·

Basic reporting

No comment.

Experimental design

No comment.

Validity of the findings

No comment.

Additional comments

The Authors did a nice work in reviewing the manuscript. I find it significantly improved.
I have no further comments.

Reviewer 2 ·

Basic reporting

The authors replied to all my comments, I have no more comments.

Experimental design

The authors replied to all my comments, I have no more comments.

Validity of the findings

The authors replied to all my comments, I have no more comments.

Additional comments

The authors replied to all my comments, I have no more comments.